# Helical Foldamers and Stapled Peptides as New Modalities in Drug Discovery: Modulators of Protein-Protein Interactions

**Keisuke Tsuchiya** [1,2]**, Takashi Kurohara** [1]**, Kiyoshi Fukuhara** [2]**, Takashi Misawa** [1,*] **and Yosuke Demizu** [1,3,4,*]

1   Division of Organic Chemistry, National Institute of Health Sciences, 3-25-26, Tonomachi, Kawasaki, Kanagawa 210-9501, Japan; p19940324@pharm.showa-u.ac.jp (K.T.); tks-kurohara@nihs.go.jp (T.K.)
2   Graduate School of Pharmacy, Showa University, 1-5-8, Hatanodai, Shinagawa-ku, Tokyo 142-0064, Japan; fukuhara@pharm.showa-u.ac.jp
3   Graduate School of Medical Life Science, Yokohama City University, 1-7-29, Yokohama, Kanagawa 230-0045, Japan
4   Graduate School of Medicine, Dentistry and Pharmaceutical Sciences, Okayama University, 1-1-1 Tsushima-naka, Kita-ku, Okayama 700-8530, Japan
*   Correspondence: misawa@nihs.go.jp (T.M.); demizu@nihs.go.jp (Y.D.); Tel.: +81-44-270-6580 (T.M.); +81-44-270-6578 (Y.D.)

**Abstract:** A "foldamer" is an artificial oligomeric molecule with a regular secondary or tertiary structure consisting of various building blocks. A "stapled peptide" is a peptide with stabilized secondary structures, in particular, helical structures by intramolecular covalent side-chain cross-linking. Helical foldamers and stapled peptides are potential drug candidates that can target protein-protein interactions because they enable multipoint molecular recognition, which is difficult to achieve with low-molecular-weight compounds. This mini-review describes a variety of peptide-based foldamers and stapled peptides with a view to their applications in drug discovery, including our recent progress.

**Keywords:** foldamers; protein-protein interaction; helical structure; building blocks; drug discovery; modality

## 1. Introduction

A foldamer, meaning "any polymer with a strong tendency to adopt a specific compact formation," is an artificial oligomer constructed from small building blocks such as amino acids, nucleic acids, and sugars [1,2]. Moreover, Hills also defined a foldamer as "any oligomer that folds into a conformationally ordered state in solution" [3]. Foldamers have been broadly used in organocatalysis [4,5], materials chemistry [6], polymer chemistry [7], chemical biology [8,9], and medicinal chemistry [10,11] because of their unique structural properties and adaptabilities [12,13]. In particular, it has been clearly demonstrated that foldamers mimic biomacromolecules and are resistant to degradation by digestive enzymes [13]. Therefore, foldamers can be potentially used in drug discovery, and many studies using foldamers have been reported [14]; among them, peptide-based foldamers have been studied extensively as possible therapeutics [15].

In general, short peptides consisting of only natural amino acids have particular problems in biological applications: (1) unstable secondary structures [16]; (2) low resistance against hydrolytic enzymes [17]; and (3) lack of cell-membrane permeability [18]. Gellman and co-workers circumvented these issues by realizing that $\alpha/\beta$-oligopeptides with a combination of $\alpha$- and $\beta$-amino acids were capable of forming stable helical structures similar to the $\alpha$-helix [1]. In addition, those $\alpha/\beta$-oligopeptides showed increased affinity against target proteins and resistance to hydrolytic enzymes [13]. Since Gellman's findings, several peptide-based foldamers have been developed that contain $\beta$-amino acids, $\gamma$-amino acids [1,3], quinoline monomers [12], and $\alpha,\alpha$-disubstituted amino acids [19,20] as building

blocks. Currently, peptide-based foldamers have been used to develop bioactive peptides such as cell-penetrating peptides (CPPs) [9], antimicrobial peptides (AMPs) [20], drug delivery systems (DDS) [21], and protein-protein interaction (PPI) inhibitors. PPIs play an essential role in maintaining the homeostasis of life, and there are hundreds of thousands of proteins involved in PPIs. For example, transcription factors are regulated by interactions with transcriptional regulators and other factors to switch transcriptional activity on and off. The dysfunction of transcription factors is related to several diseases, such as various cancers, diabetes, and neurological disorders [22]. Therefore, drug discovery approaches inhibiting aberrant PPIs are important for developing therapeutics to treat these diseases. However, the interaction surface of PPIs is generally broad and flat [23], and the lack of druggable binding pockets has become a bottleneck in drug discovery research using small molecule compounds. Accordingly, peptide-based foldamers are expected to be a new modality for developing PPI inhibitors because they can target a wide interface between protein-protein interactions. Furthermore, it has been reported that many PPIs are mediated by the $\alpha$-helical and $\beta$-sheet secondary structures [24,25]. In particular, the $\alpha$-helix plays an important role in expressing diverse functions, such as recognizing DNA and other proteins. Therefore, foldamers mimicking $\alpha$-helices represent promising candidates as PPI inhibitors. In this review, we introduce past and recent research on PPI inhibitors based on helical foldamers by introducing several building blocks such as $\alpha,\alpha$-disubstituted amino acids, $\beta$- and $\gamma$-amino acids, and quinoline monomers. (Figure 1). In the same class, $\delta$- and $\varepsilon$-peptides, aza-$\beta^3$-peptides, pyrrolinones, $\alpha$-aminooxy-peptides and sugar-based peptides have been reported as foldamers and have been investigated for the development of bioactive peptides [2,26–30]. Moreover, the side-chain stapling of the peptides has been widely used to stabilize their helical structure, and many PPI inhibitors based on stapled peptides have been reported [31]. On the other hand, the PROTACs are a novel strategy for target protein degradation and are expected to be the next-generation drugs. Recently, the PROTAC based on peptide-based PPI inhibitors has been reported. In this review, the recent progress of the development of PPI inhibitors based on stapled peptides and the PROTACs using peptide-based PPI inhibitors were also introduced.

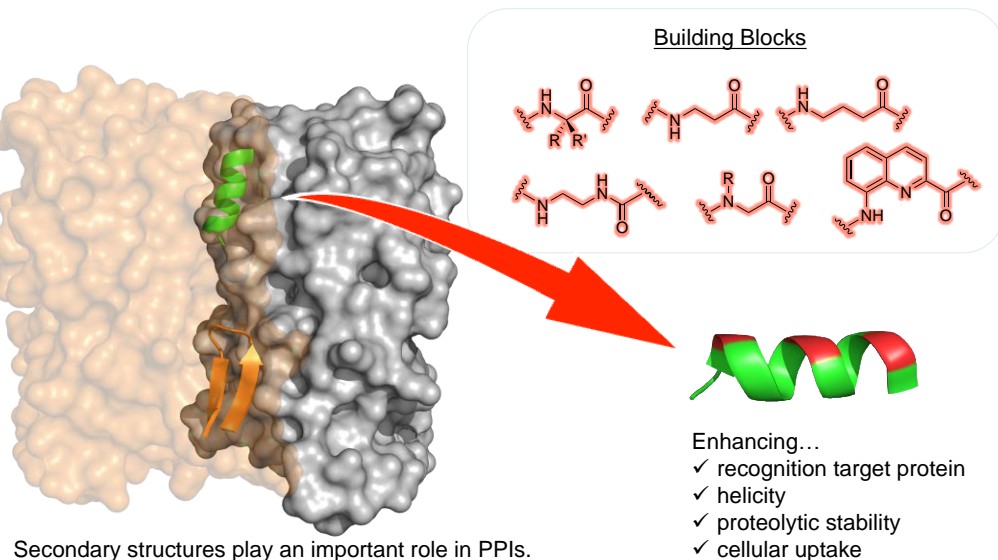

**Figure 1.** Inhibition of protein-protein interactions by peptide-based foldamers containing non-proteinogenic amino acids.

## 2. $\alpha$-Peptides

The $\alpha$-helix is a right-handed helical structure formed by hydrogen bonds between atoms that form the peptide backbone. Specifically, the $\alpha$-helix forms a hydrogen bond between the C=O of the (*i*)-position amino acid and the N–H of the (*i* + 4)-position amino

acid, resulting in a right-handed helical structure that has 3.6 residues per turn and a rise of 0.54 nm. The α-helix is involved in most PPIs and can be used as a general template for inhibitor design. Sufficient PPI inhibition requires the design of peptides that form stable helical structures. α,α-Disubstituted amino acids (dAAs), non-proteinogenic amino acids that stabilize peptide helical structures, have chemical structures in which the α-position hydrogen of natural α-amino acids is replaced by an alkyl group [32–35]. Various dAAs have been developed. dAAs can stabilize the secondary structures of oligopeptides, and α-methylated and cyclic dAAs are often used [32–38]. dAAs are incorporated into functionalized helix-stabilized peptides such as organocatalysts [38–40], drug delivery system (DDS) carriers [41–46], and antimicrobials [47–50]; however, there are only a few examples of their use as PPI inhibitors. Reported PPI inhibitors containing dAAs are listed in Table 1. 2-Aminoisobutyric acid (Aib) [51–53], the simplest dAA, is used as a helix promoter. Aib-based helical peptides have been used as PPI inhibitors targeting proteins such as mouse double minute 2 (MDM2) [54,55] and S100B [56]. We have reported that short Leu-rich peptides that incorporate a combination of Aib and a stapling side-chain with dihydroxy groups form stable helical structures. Those peptides inhibit the binding of a coactivator to the vitamin D receptor (VDR) [57]. In addition, a peptide with Aib replaced by hydroxymethylserine also formed a stable helical structure and possessed VDR-coactivator inhibitory activity [58].

**Table 1.** α-Peptide PPI inhibitors that contain dAAs.

| Peptide Sequence | Target Protein | Ref. |
|---|---|---|
|  | hDM2 | [54] |
|  | MDM2 | [55] |
|  | S100B | [56] |
|  | VDR | [57] |
|  | VDR | [58] |

## 3. β-Peptides

β-peptides are composed of β-amino acids (β-AA), which have an amino group at the β$^3$-position and the side-chain on the β$^2$- or/and β$^3$-position from the carboxylic acid. β-peptides form specific secondary structures such as 12- or 14-membered helices, depending on their sequences [59–62]. Applications in structural chemistry [63,64], materials [6,65], and organocatalysis [66,67] have been carried out using β-peptides because these peptides have unique structural properties. β-peptides show tolerance against digestive enzymes in the body, and thus, β-peptides represent a platform for developing bioactive peptides [68]. However, the 12- and 14-membered helices of β-peptides are not readily recognized by biomacromolecules such as nucleic acids and proteins because these helices differ in structure to that of the α-helix, and applications of β-peptides in drug discovery have been limited. Based on this background, Gellman and co-workers have reported that the introduction of β-AA into α-peptides with a specific pattern can mimic the native α-helical structure. That is, α/β-peptides that exhibit the −ααβααβ pattern, in which the β-AA residues align in a "stripe" along one side of the helix, can support functional α-helix mimicry [69]. In contrast, for α/β-peptides that display the ααβ or αααβ pattern, the β-AA residues spiral around the periphery of the helix [69]. Furthermore, Gellman and co-workers developed cyclic β-amino acid residues, such as trans-2-aminocyclohexane carboxylic acid, (*R,R*)-*trans*-2-aminocyclopentanecarboxylic acid (ACPC), and (3*S*,4*R*)-*trans*-3-aminopyrrolidine-4-carboxylic acid (APC), and the introduction of these cyclic β-amino acids into peptides stabilized helical structures and enhanced their biological activity [70]. Their efforts showed that combining α- and β- amino acids is a promising strategy for developing bioactive peptides. Based on this knowledge, several cyclic and bicyclic β-AAs have been reported and applied for the development of bioactive α/β-peptides [71,72]. Currently, β-peptides are studied widely for several bioactive peptides, such as CPPs [19,73], AMPs [74,75], and PPI inhibitors [76–81]. Here, recent advances in β-peptide-based PPI inhibitors are described (Table 2).

Schepartz and co-workers designed and synthesized β-peptides containing β$^3$-amino acid residues and analyzed their secondary structures [76,77]. In addition, the PPI inhibitory activities targeting the p53/hDM2 interaction, which is a transcriptional activator critical for stress-induced cell cycle arrest and apoptosis, were evaluated. Cancer cells often overexpress hDM2 and downregulate p53, resulting in the promotion of cell proliferation. Therefore, peptides that inhibit the interaction between p53 and hDM2 are promising candidates for cancer therapy. The β-peptides designed using the binding motif of p53 against hDM2 and β$^3$-AA formed stable 14-membered helical structures via a salt bridge between β$^3$-homoglutamate and β$^3$-homoornithine residues. Binding affinity determined using the fluorescein polarization assay revealed that the β-peptides showed higher affinity toward hDM2 when compared with the affinity of the parent α-peptides toward hDM2. They also reported the β-peptides as scaffolds for developing PPI inhibitors that block human immunodeficiency virus-1 (HIV-1) fusion using the same strategy [78]. As described above, the introduction of β-AA residues in specific patterns should mimic the α-helix [69]. Based on this knowledge, α/β-peptides containing cyclic β-AA residues based on the C-terminal heptad-repeat (CHR) domain of HIV protein gp41 were designed [79]. gp41 is expressed on the surface of the envelope of HIV and forms a critical bundle intermediate that drives the fusion of the viral envelope with the target cell membrane. The formation of bundle intermediates requires the interaction of the CHR with the N-terminal heptad repeat (NHR) domain. Therefore, CHR mimics may inhibit the formation of bundles and inhibit the fusion process of HIV. α/β-Peptides with the ααβααβ pattern were designed to stabilize the helix structure through the formation of a salt bridge. The α/β-peptides showed higher affinity against the NHR domain than the parent α-peptide. Moreover, the introduction of β-AAs into the sequence prolonged the half-life of the peptides against digestive enzymes by 1000-fold. Thus, α/β-peptides increased the affinity toward target proteins and improved peptide stability, demonstrating the potential development of therapeutic reagents.

**Table 2.** β-Peptides as PPI inhibitors.

| Peptide Sequence and Structure | Target PPI | Ref. |
|---|---|---|
| H-β³O-β³V-β³W-β³E-β³V-β³W-β³O-β³V-β³I-β³E-OH <br><br> R / $H_2N$ / $CO_2H$ / **β³-AA** | HIV gp41 | [77] |
| H-β³K-β³V-β³L-β³E-β³V-β³W-β³K-β³V-β³F-β³E-OH | p53-hDM2 | [78] |
| Ac-(β³R)-TWE-(β³E)-WD-(β³R)-AIA-(β³E)-YA-(β³R)-RIE-(β³E)-LI_Z_AAQ-(β³E)-QQ_Z_KNE-(β³E)-AL_Z_EL-NH₂ <br><br> $H_2N$ / COOH / **APC (_Z_)** | HIV gp41 | [79] |
| H-V-(β³D)-NK-(β³F)-NKE_X_CN_Z_RAIE_U_ALDPNLND_U_QFH_U_KIW_Z_IK_X_DC-NH₂ <br><br> $H_2N$ / COOH / **ACPC (_X_)** | VEGF | [80] |

Gellman and co-workers developed α/β-peptides that target the diverse protein-protein interaction interface, which may be an alternative to antibodies, with high selectivity and affinity against target proteins such as vascular endothelial growth factor (VEGF) [80,81]. VEGF is a homodimeric protein that binds to the VEGF receptor to trigger intracellular signaling for angiogenesis. In cancer cells, VEGF is overexpressed and promotes angiogenesis, causing an exacerbation of cancer. Therefore, inhibitors of VEGF/VEGF receptor signaling are promising reagents for cancer therapy. Gellman and co-workers designed α/β-peptides based on the three-helix bundle Z-domain scaffold, which was developed via phage display targeting the VEGF monomer. The Z-domain targeting VEGF (Z-VEGF) is composed of three helices, with helices 1 and 2 associated with binding to the surface of VEGF and helix 3 stabilizing the entire structure of Z-VEGF. Helix 3 was removed because it positions on the opposing side of the binding interface, and the open side of helices 1 and 2 was replaced with β-AAs to stabilize the structures. The designed α/β-peptides exhibited high affinity toward VEGF with affinities similar to that of the parent α-peptide. The α/β-peptides showed anti-proliferation activity against HUVEC cells, indicating that the α/β-peptides antagonize VEGF/VEGF receptor signaling. Thus, the introduction of β-AAs can enhance bioactivity and improve pharmacokinetics, and α/β-peptide foldamers represent potential novel drug modalities that target broad interfaces of PPIs.

## 4. γ-Peptides

γ-amino acids (γ-AAs) have an amino group at the γ-position relative to the carboxyl group, and γ-peptides adopt ordered secondary structures, as observed for β-peptides [82]. Currently, *cis*-γ-amino-proline [83,84], vinyl type γ-AAs [85,86], and cyclic γ-AAs [87] have been used widely as building blocks of γ-peptides. Some types of γ-peptides containing γ-substituted, α,γ-substituted, or α,β,γ-substituted γ-AAs adopt 12- or 14-membered helical structures [88,89]. Gellman and co-workers reported that α,β,γ-peptides containing β- and γ-AAs in the αγααβα repeat formed α-helical structures in an aqueous solution [90]. Their

findings revealed that $\alpha/\beta/\gamma$-peptides mimic the $\alpha$-helix and are suitable for designing bioactive peptides. In this section, PPI inhibitors containing $\gamma$-AA are described (Table 3). Aitken and co-workers reported that $\alpha/\beta/\gamma$-peptides composed of $\alpha$- and $\gamma$-AAs and *trans*-2-aminocyclobutanoic acid (tACBC) form 12 or 13-membered helices [91,92]. A series of $\alpha/\beta/\gamma$-peptide derivatives were designed as potential bioactive peptides based on their peptide design. The designated $\alpha/\beta/\gamma$-peptides bound hDM2 to inhibit the interaction between hDM2 and p53 and showed high tolerance against digestive enzymes. Cai and co-workers reported that a sulfono-$\gamma$-peptide, which is an oligomer of the *N*-acylated-*N*-aminoethyl amino acid, can adopt a helical structure bearing a pitch of 5.1 Å, which is similar to that of the $\alpha$-helix (5.4 Å) [93,94]. The sulfono-$\gamma$-peptides present side-chains similarly to the side-chain presentation found in $\alpha$-helices by introducing proper side-chains on the sulfone groups and the $\gamma$-positions. Based on these structural properties, PPI inhibitors targeting the $\beta$-catenin/BCL-9 interaction were developed [95]. The competition assay revealed that the designed sulfono-$\gamma$-peptides disrupted the $\beta$-catenin/BCL-9 interaction with an $IC_{50}$ value of 0.74 µM. Furthermore, the sulfono-$\gamma$-peptides were internalized effectively into SW480 cells and inhibited Wnt/$\beta$-catenin signaling by disrupting the $\beta$-catenin/BCL-9 interaction. This research group also developed PPI inhibitors against the p53-MDM2/MDMX interaction [96]. The sulfono-$\gamma$-peptides mimic an $\alpha$-helix and effectively penetrate the cell membrane [95]. Thus, the sulfono-$\gamma$-peptides are promising tools for developing intracellular PPI inhibitors. Moreover, Maillard et al. reported that the $\gamma$-peptides, composed of thiazole-based $\gamma$-AAs, adopted the 9-helix and interaction with amyloid-$\beta$ peptides [97,98]. Their efforts revealed that the thiazole-based $\gamma$-AAs could be applicable for the development of PPI inhibitors by mimicking the helical structures.

**Table 3.** $\gamma$-peptides as PPI inhibitors.

| Peptide Sequence and Structure | Target PPI | Ref. |
|:---:|:---:|:---:|
| Boc-F-($\gamma^4$A)-(tACBC)-($\gamma^4$W)-(tACBC)-($\beta^3$L)-OMe  | p53-*h*DM2 | [91] |
| Ac-$\gamma^4$E$^{(EtNH2)}$-$\gamma^4$R$^{(Me)}$-$\gamma^4$R$^{(Me)}$-$\gamma^4$E$^{(Me)}$-$\gamma^4$L$^{(EtNH2)}$-$\gamma^4$T$^{(iBu)}$-$\gamma^4$L$^{(EtNH2)}$-$\gamma^4$R$^{(iBu)}$- $\gamma^4$A$^{(Me)}$-$\gamma^4$L$^{(iBu)}$-$\gamma^4$L$^{(tol)}$-NH$_2$  | $\beta$-catenin/BCL9 | [95] |
| Ac-$\gamma^4$L$^{(p\text{-}ClPh)}$-$\gamma^4$F$^{(Me)}$-$\gamma^4$E$^{(Me)}$-$\gamma^4$W$^{(Me)}$-$\gamma^4$K$^{(Me)}$-$\gamma^4$cBA$^{(iBu)}$-$\gamma^4$A$^{(Me)}$-NH$_2$ | P53-MDM2/MDMX | [96] |
|  | Amyloid-$\beta$ | [98] |

### 5. Peptoids

*N*-substituted polyglycines as peptide mimetics are referred to as peptoids [99]. Peptoids are also classified as foldamers that aim to enhance proteolytic stability while mimicking α-helical and β-sheet secondary structures. The proton on the amide nitrogen of the backbone is replaced by an alkyl substituent in peptoids. Conventional peptoids do not have asymmetric carbons on the main chain and cannot form intramolecular hydrogen bonds because they have no amide protons. Therefore, those peptoids are typically flexible and have reduced polarity. Peptoids are attracting attention for their use as intracellularly targeted PPI inhibitors because they are easy to synthesize and display high membrane permeability as they do not form hydrogen bonds with water. In particular, applying peptoids as PPI inhibitors has been attempted by using chemical modifications that allow these peptoids to form stable secondary structures (Table 4). Peptoids with chiral substituted groups can form well-defined one-handed helical structures [100–103]. Furthermore, several studies have reported recently that the conformations of peptoids can be constrained by adding a methyl group to an α-carbon atom [104,105]. Peptoids have been applied for drug discovery, such as vaccine therapeutics [102] and PPI inhibitors. From combinatorial libraries, vascular endothelial growth factor receptor2 (VEGFR2)-binding peptoids have been screened, and peptoid antagonists that exhibit VEGFR2 activity in vitro and in vivo have been developed [106–109]. Kirshenbaum and co-workers have successfully created in silico screening of PPI inhibitors using a rigid cyclic peptoid skeleton [110]. Kodadek [111] and Lim [112] groups have individually succeeded in obtaining inhibitors of intracellular proteins from combinatorial libraries of peptoids. In addition, Morimoto and Sando's group recently succeeded in preparing peptoid foldamers with oligo-*N*-substituted alanine (oligoNSA) that stably form specific conformations by introducing asymmetric substituents on the peptoid backbone [104]. Furthermore, this oligoNSA is a useful framework for intracellular PPI inhibitors [113]. Foldamers with combinations of α-amino acids and peptoid monomers are also potent PPI inhibitors. Peptide-peptoid hybrids have been reported to be potent and highly selective ligands for the Grb2 SH3 domain [114]. As another example, peptide-peptoid hybrids that form stable β-hairpin structures were reported to inhibit CXCR4-mediated HIV entry [115]. Peptoid-based (i.e., foldamers) drug discovery targeting intracellular PPIs will likely be realized using various approaches in the future.

**Table 4.** Peptoid-based PPI inhibitors.

| Sequence or Structure | Target Protein | Ref. |
|---|---|---|
|  | VEGFR2 | [106] |
|  | β-catenin | [110] |

**Table 4.** *Cont.*

| Sequence or Structure | Target Protein | Ref. |
| --- | --- | --- |
|  | Rpn13 | [111] |
|  | Skp2 | [112] |
|  | MDM2 | [113] |
| YEVPPPV<u>X</u>PRRR <br>  <br> X | Grb2 SH3 | [114] |
|  | CXCR4 | [115] |

## 6. Urea-Type Foldamers

Urea-type helical foldamers are classified as a type of γ-peptide in which a methylamino motif is inserted into α-amino acids and have been used in drug discovery research [116,117]. In 1995, a solid-phase synthetic method was reported by Burgess et al. [118,119]. In their

method, the urea bond was constructed from isocyanate prepared from the carboxyl group of natural amino acids. In 2000, Guichard and co-workers developed a building block with an hydroxysuccinimide group as the reactive functional group for polyurea synthesis (Figure 2) [120]. This building block is currently used as the primary synthetic method for polyurea construction. The urea building blocks are derived from natural amino acids. Therefore, their side-chain environments closely resemble those of natural helical peptides. The characteristic features of polyurea peptides, studied extensively by Guichard and co-workers, include forming helical structures with 2.5 residues per turn via intramolecular hydrogen bonds [121,122]. This structural feature is found in native peptides, which form helical structures by hydrogen bonding between amide and carbonyl groups in the peptide backbone, indicating that polyurea foldamers can be designed to mimic natural peptides.

Peptide

Urea foldamer

**Figure 2.** Comparison of the $\alpha$-peptide and urea foldamer.

Several applications using urea-type helical foldamers have been reported. As listed in Table 5, in 2019, Goudreau, Guichard, and co-workers reported that urea-based foldamers could be used for designing a glucagon-like peptide 1 (GLP-1) analog [123]. GLP-1 is a physiological peptide secreted from L-cells in the small intestine, which binds to the GLP-1 receptor on pancreatic beta-cells and stimulates insulin secretion. In their report, the urea-peptide, which replaces residues 14–21 of GLP1 with $Y^uE^uA^uA^uA^uA^u$, exhibited high degradation stability in mouse plasma while maintaining its blood glucose inhibitory effect in mice. Enhanced metabolic stability is an important factor in drug discovery, as well as the stability of the helical structure. Urea-containing PPI inhibitors targeting the transcriptional regulators MDM2 and VDR have also been reported by Guichard and co-workers in 2021 [124]. The ubiquitin ligase MDM2 negatively regulates the tumor suppressor p53. Thus, the inhibition of MDM2 restores p53 and inhibits cancer cell growth, and the PPI inhibitors of MDM2 represent potential anticancer agents [125]. The PMI peptide consisting of 12 residues (TSFAEYWNLLSP) that binds to MDM2 was found by phage display analysis. Guichard and co-workers focused on the ten residues sequence TSFAEYWNLL of PMI and designed the corresponding urea peptide $TSFAEYW^uA^uL^uA^u$. As for peptide design, Wu was introduced at the seventh position to improve protease resistance, and $A^uL^uA^u$

was introduced to induce helix formation. This urea-type peptide binds to MDM2 with high affinity. In addition, a urea-type peptide based on the structure of SRC1-2 and SRC2-3 tridecapeptides consisting of a central consensus LXXLL motif was designed to inhibit VDR, and coactivator interactions were also developed. VDR, a nuclear hormone receptor (NHR), is associated with regulating many biological functions such as bone homeostasis, cell growth, and immunity. For VDR to transcribe 1,25-dihydroxyvitamin D3 [1,25-(OH)$_2$D$_3$] as a ligand, molecular binding of the ligand-binding domain (LBD) to coactivators such as steroid receptor coactivators (SRC1, SRC2, SRC3) is required. Therefore, inhibiting the interaction between VDR and coactivators is a potential therapeutic strategy for Paget's disease of bone [126]. Guichard and co-workers designed a hexa-urea (A$^u$A$^{u\alpha}$L$^u$R$^u$L$^u$Nle$^u$KDD) peptide in which residues 1–10 of SRC2–3 (ENALLRYLLDKDD) were replaced, and the urea-peptide showed a 10-times stronger binding affinity to VDR ($K_d$ = 0.14 μM) compared with SRC2–3 ($K_d$ = 1.5 μM). A urea-foldamer PPI inhibitor targeting anti-silencing function 1 (ASF1) has also been reported [127]. ASF1 is a histone H3/H4 chaperone that assembles and disassembles chromatin during transcription, replication, and repair. ASF1 dysfunction is associated with various pathologies, including age-related diseases, pathogen infections, and cancer. Depleting ASF1 inhibits the growth of various cancer cell lines and enhances the sensitization of cells to chemotherapeutic agents [128]. An ASF1 inhibiting urea-peptide (Ac-EKNal$^u$R$^u$L$^u$Q$^u$RIA-NH$_2$) with four urea residues inserted in the helix center based on the C-terminal sequence of H3 (ASTEEKWARLARRIAGAGGVTLDGFG), which forms the interaction interface between ASF1 and H3, was designed. The urea-peptide showed high stability against proteolysis under stringent conditions and demonstrated binding to ASF1 at low micromolar values ($K_d$ = 2.7 μM). In addition, the X-ray co-crystal structure of the urea-peptide and ASF1 was also analyzed. Those results are expected to contribute strongly to the future development of ASF-targeted drug discovery.

**Table 5.** Urea-based foldamers and their applications.

| Structures and Sequences | Target | Ref. |
|---|---|---|
| H-HGEGTFTSDVSSY Y$^u$E$^u$A$^u$A$^u$A$^u$A$^u$ FIAWLVKGRG-NH$_2$ | GLP-1 mimic | [123] |
| TSFAEYW$^u$A$^u$L$^u$A$^u$ | MDM2 | |
| A$^u$A$^{u\alpha}$L$^u$R$^u$L$^u$Nle$^u$KDD  * A$^{u\alpha}$ = | VDR | [124] |
| Ac-EKNal$^u$R$^u$L$^u$Q$^u$RIA-NH$_2$ <br> * Nal: Naphthylalanine | ASF1 | [127] |

As summarized above, replacement with three to four urea residues can improve the metabolic stability of PPIs without significantly affecting their inhibitory activity. These studies will contribute to the future development of PPI drug discovery based on urea-foldamers.

## 7. Aromatic Foldamers and the Terphenyl Scaffold

Helical peptide foldamers containing aromatic building blocks have been developed (Figure 3). Aromatic peptide foldamers generally adopt helical structures similar to helicene [117,129]. The driving force stabilizing the helical structures is the π-π stacking interaction between aromatic rings and hydrogen bonding between the ring nitrogen lone pairs and amide bonds. In 1994, Hamilton and co-workers initially reported aromatic foldamers composed of anthranilic acid amide [130,131] and pyridine-2,6-dicarboxylic acid [129,131]. Subsequently, in 2003, quinoline-type [132,133] foldamers were reported by Huc and co-workers. Ortho-terphenyl is another motif used commonly to construct

helical structures, and its peptide-type helical foldamers have been reported by Gellman and co-workers recently [134].

**Figure 3.** Structures of aromatic foldamers.

As listed in Table 6, in 2019, Huc and co-workers demonstrated the binding and immobilization of quinoline-type helical foldamers against two cysteine mutant proteins, interleukin 4 (IL4) and cyclophilin A (CypA), based on a tethering approach [135]. A cysteine-reactive probe, 2-(1$H$)-thiopyridone, was introduced into the N-terminus of the quinoline-based helical aromatic peptide ($Q^{Leu}Q^{Leu}Q^{Orn}Q^{Asp}$). This Cys-reactive helical peptide was designed to react with the SH group near the PPI site by an S-S exchange reaction. Such covalent interactions have been demonstrated in experiments using site-directed Cys mutants of CypA and IL4. This technology is expected to be applied to design PPI foldamers targeting Cys residues and contribute to the structural elucidation of PPIs. In 2018, Huc and co-workers synthesized a unique oligoamide-based foldamer that mimics the negatively charged phosphate site of B-DNA and takes on a single-helix structure [136]. These mimics disrupted the activity of DNA-interacting proteins targeted for cancer therapy and exerted their cytotoxicity only in the presence of transfection agents. Furthermore, in 2021, this phosphorylated quinoline foldamer was conjugated as a payload to trastuzumab for selective transport to breast cancer cells [137]. These phosphorylated foldamers are also expected to be used to develop PPI inhibitors.

**Table 6.** Aromatic foldamers and their applications.

| Structures and Sequences | Target | Ref. |
|---|---|---|
| X-Q$^{Leu}$Q$^{Leu}$Q$^{Orn}$Q$^{Asp}$ <br> Q$^{Leu}$: R = (isobutyl ether) <br> Q$^{Orn}$: R = (propylamine ether, NH$_2$) <br> Q$^{Asp}$: R = (ether CH$_2$CO$_2$H) <br> X = (2-pyridyl disulfide ketone) | CypA, IL4 | [135] |
| -[Q$^{Pho}$]$_{16}$- <br> (phosphonomethyl ether quinoline carboxamide) | HER2 | [137] |

## 8. Stapled Peptides

In recent years, side-chain intra-cross-linking of peptides has received attention as a novel strategy for regulating the secondary structures of peptides. This section introduces cross-linked peptides that contribute to the potential development of PPI inhibitors based on the stabilization of α-helical structures (Table 7). Methods for stabilizing α-helix peptide folds have been developed based on interactions between side-chains of natural amino acids within peptide sequences. In 1987, Marqusee and Baldwin revealed that α-helical structures are stabilized by intramolecular salt bridges formed between the side-chains of lysine and glutamic acid residues at the *i* and *i* + 4 positions [138]. Chelation of metal ions using iminodiacetic acid-bearing side-chains or histidine residues on peptides has been reported to stabilize the peptide α-helix [139,140]. Moreover, in 2013, Tezcan and co-workers revealed that α-helix stabilization is achieved by coordinating a divalent metal ion between a histidine residue and an 8-hydroxyquinoline moiety [141].

Subsequently, methods have been developed to stabilize α-helical structures by forming intramolecular covalent side-chain cross-linking, such as the disulfide bond or lactam bridge. Burris and co-workers reported PPI inhibitors that formed stable α-helices through the formation of disulfide bonds. [142,143]. This disulfide-bond-containing peptide, the peptidomimetic estrogen receptor modulator 3 (PERM3), inhibited interactions between estrogen receptor α (ERα) and its coactivator [142,143]. Our group reported that the conjugation of CPPs, which penetrate the cell membrane and deliver cargo molecules into cells, with PERM3 enhanced the intracellular delivery of PERM3 peptides and inhibited ERα gene expression in mammalian cells [144]. Pei and co-workers reported that α-helical peptides containing a lactam bridge between Asp and Lys inhibited the p53/ MDM2 interaction [145]. The interaction between p53 and MDM2 plays a vital role in the homeostasis of mammalian cells, and overexpression of MDM2 reduces the expression level of p53, resulting in the exacerbation of cancers. The efforts by Pei and co-workers revealed that the formation of a lactam bridge between Asp and Lys residues stabilized the α-helical structure and inhibited the PPI between p53 and MDM2, thus hampering cell growth of tumor cells.

Side-chain cross-linking of peptides using not only natural amino acids but also non-proteinogenic amino acids has also been developed, which is called "stapling" [146]. In particular, hydrocarbon stapling has been used to develop PPI inhibitors that target various proteins. Grubbs and co-workers developed stapling between L-serine or L-homoserine *O*-allyl ethers using the olefine metathesis reaction for α-helical stabilization of oligopep-

tides [147]. Our group reported that the stapled peptide, DPI-07, which introduced a side-chain cross-link between serine and homoserine *O*-allyl ether, inhibited the interaction between VDR and its coactivator [57].

In 2000, Verdine and co-workers designed and synthesized α-methylated amino acid with an olefin moiety of different lengths and investigated the effects of the position and the length of the olefinic side-chain on cross-linking formation and peptide secondary structures [148]. In this report, introducing hydrocarbon cross-linking at the *i* and *i* + 4 positions using two (*S*)-4-pentenylalanines or at the *i* and *i* + 7 positions using (*R*)-7-octenylalanine and (*S*)-4-pentenylalanine enhanced the biological compatibility of peptides, such as helicity and resistance to hydrolysis [146,148]. In addition, introducing hydrocarbon cross-linking into peptides enhanced their molecular recognition and cellular uptake [149]. Verdine and co-workers reported a stapled peptide at the *i* and *i* + 4 positions and showed that this peptide, 35R, inhibited the interaction between β-catenin and T cell factor (TCF). The peptide 35R showed a strong binding affinity at nanomolar levels against β-catenin and downregulated target gene expression in mammalian cells [150]. Sawyer and co-workers revealed that ATSP-7041 (ALRN-6924), a peptide with hydrocarbon cross-linking at the *i* and *i* + 7 positions, inhibited the function of MDM2 and murine double minute X (MDMX). Moreover, ATSP-7041 has been shown to have anti-tumor activity in vivo [151] and is currently under clinical trials [152]. In 2021, Ueda and co-workers revealed that jasmonate-ZIM-domain (JAZ)-derived peptides bearing double staple moieties, JAZ9, inhibited one of the transcriptional factors of MYC in the jasmonate signaling pathway of plant cells [153]. In 2014, Verdine and co-workers reported a stitched peptide that formed a bis-hydrocarbon cross-linking using (*S*)-7-octenylalanine at the *i* and *i* + 7 positions and (*S*)-4-pentenylalanine at the *i* and *i* + 4 positions via α,α-bis-4-pentenylglycine. The stitched peptide displayed enhanced bio-acceptability, such as thermal and α-helical stability, proteolytic resistance, and cellular uptake [154]. In 2020, Partridge and co-workers reported that a stitched peptide containing D-amino acids inhibited the interaction between p53 and MDM2/MDM4 at the cellular level [155].

Other side-chain cross-links, such as azobenzene, the aryl staple, Ugi stapling, and the hydrogen bond surrogate, have been reported to stabilize α-helical structures of peptides. In 2002, Woolley and co-workers reported controlling the secondary structure of short oligopeptides with an azobenzene moiety, which undergoes *cis-trans* transitions upon irradiation at specific wavelengths [156]. In 2019, Hamm and co-workers revealed that the azobenzene-type staple controls the binding activity of the S-peptide, which is composed of RNase S, by conformational changes upon UV irradiation [157]. Pentelute and co-workers reported α-helix-stabilized peptides with perfluoroaromatic cross-linking of two cysteine residues at the *i* and *i* + 4 positions by the nucleophilic aromatic substitution ($S_N Ar$) reaction. This method can be applied to unprotected peptides, and peptides modified by this cross-linker displayed higher binding affinity toward target proteins, enhanced proteolytic stability, and greater cellular uptake [158]. In 2019, Rivera and co-workers reported lactam-type stapled peptides using the on-resin Ugi reaction with formaldehyde and a cyanide component. The introduction of cross-links into peptides using the Ugi reaction increases the stability of the α-helix fold and facilitates the modification of relevant fragments such as sugar, lipids, and fluorescent labels on the stapling moiety [159]. This method can be used to develop functionalized peptides for DDS and chemical biology. For example, stapled peptides with high binding affinities toward MDM2 and MDMX were developed using this method [160]. A method to form cross-linking in the peptide backbone has also been developed. The hydrogen bond surrogate (HBS) is a peptide with the intramolecular hydrogen bond in the backbone at the *N*-terminal's *i* and *i* + 4 positions replaced with a covalent carbon-carbon bond through an olefin metathesis reaction [161]. HBS-type peptides stabilize α-helices in short sequences [161]. In addition, these peptides are expected to have improved biological features over native peptides, such as proteolytic stability, molecular recognition, and cellular uptake [161]. An HBS-type peptide has been shown to exhibit binding activity in regions completely buried with a protein [162].

Recently, protein degraders called proteolysis-targeting chimeras (PROTACs) have attracted significant attention in developing therapeutic reagents. Among them, PPI ligands have been used to develop PROTACs. PROTACs induce targeted protein degradation via the ubiquitin-proteasome system (UPS), which involves poly-ubiquitination of the target protein (POI) by forming a complex with ubiquitin ligase (E3) [163]. In this method, a chimeric molecule consisting of an E3 ligand, a POI ligand, and a conjugating linker binds each protein to form a ternary complex [163]. Subsequently, the E3 ligase forcibly ubiquitinates the POI to induce degradation [163]. Von Hippel–Lindau (VHL), Cereblon, and inhibitor of apoptosis protein (IAP) as E3 ligases are often referred to as PROTACs. Please refer to reviews on PROTACs [163,164] for a detailed explanation. In recent years, peptide-based PROTACs have been developed. As representatives, Chen and co-workers reported xStAx peptides developed by the Verdine group [150] against β-catenin, and the xStAx-based PROTAC showed degradation activity against β-catenin [165]. Our group developed the stapled peptide (SAHM1 [166])-based protein degraders that target Notch [167]. Furthermore, the stapled peptide-based LCL-stPERML-R7 targeting ERα has been reported recently by our group [168]. Thus, structurally controlled peptides can be used as PPI inhibitors and PROTAC ligands, which are expected to be a new modality.

**Table 7.** Stapled PPI inhibitor peptides.

| Peptide Sequence and Structure | Target Protein | Ref. |
|---|---|---|
| <br>Npg: Neopentylglycine<br>Lowercase letters: D-amino acid | ERα | [143] |
|  | MDM2/p53 | [145] |

**Table 7.** *Cont.*

| Peptide Sequence and Structure | Target Protein | Ref. |
|---|---|---|
|  | VDR | [57] |
|  Ac-RRWPR-$S_5$-ILD-$S_5$-HVRRVWR-NH$_2$ | β-catenin/TCF | [150] |
|  Ac-LTF-$R_8$-EYWAQ-Cba-$S_5$-SAA-NH$_2$  Cba | MDM2/p53 MDMX/p53 | [151] |
|  TMR-SVP-$S_5$-ARK-$S_5$-SL-$S_5$-RFL-$S_5$-KRKERL-OH  TMR: Tetramethylrhodamine | MYC | [153] |

**Table 7.** *Cont.*

| Peptide Sequence and Structure | Target Protein | Ref. |
| --- | --- | --- |
|  H-R$_5$-a-6-F-w-y-B$_5$-n-p-CF$_3$-f-ekll-R$_8$-NH$_2$ <br> 6-F-w    p-CF$_3$-f | MDM2/p53 MDM4/p53 | [155] |
|  KETAACKFERQHCDSSTSAA <br> X: 3,3′-bis(sulfonato)-4,4′-bis-(chloroacetamido)azobenzene | S-protein | [157] |
|  H-ITFCDLLCYYGKKK-NH$_2$ <br> X: ... or ... | C-terminal domain of an HIV-1 capsid assembly polyprotein (C-CA) | [158] |
|  Ac-LTFEQYWAQLESAA-NH$_2$ | MDM2/p53 MDMX/p53 | [160] |

**Table 7.** *Cont.*

| Peptide Sequence and Structure | Target Protein | Ref. |
|---|---|---|
| X-NI-†-SLLRVQAHIRKKMV-NH$_2$ <br><br> X-NI-† | myoA tail interacting protein (MTIP) | [162] |
| (LCL-Linker)-KILR-S$_5$-LLQ-S$_5$-GGGRRRRRRR-NH$_2$ <br><br> LCL-Linker | ERα | [168] |

## 9. Outlook

In this mini-review, we have introduced research on peptide-based foldamers composed of several building blocks and their applications as PPI inhibitors. Peptide-based foldamers consisting of various building blocks, such as α-AAs, β-AAs, γ-AAs, side-chain cross-linking, peptoids, and other non-proteinogenic amino acids, form stable helical structures with high affinities toward target proteins. Although foldamers represent potential drugs, challenges such as cell-membrane permeability and specific tissue targeting must be resolved to realize their full therapeutic potential. Recent research on CPPs revealed that increasing the stability of the secondary structure enhances intracellular uptake, indicating that stable helical foldamers may overcome issues associated with membrane permeability. Future development of new building blocks and further optimization of foldamer stability should yield therapeutics that target intracellular PPIs associated with various diseases. Thus, peptide foldamers functioning as PPI inhibitors are expected to be developed in future drug discovery platforms.

**Author Contributions:** Writing—original draft preparation, K.T., T.K., T.M. and Y.D.; writing—review and editing, K.F., T.M. and Y.D.; supervision, T.M. and Y.D.; funding acquisition, Y.D. All authors have read and agreed to the published version of the manuscript.

**Funding:** This study was supported in part by grants from the Japan Agency for Medical Research and Development (22mk0101197j0002, 22fk0210110j0401, 22ak0101185j0301, and 22fk0310506j0701 to Y.D.), Japan Society for the Promotion of Science and the Ministry of Education, Culture, Sports, Science and Technology (JSPS/MEXT KAKENHI Grants Number JP18H05502 and 21K05320 to Y.D.; 20K06958 to T.M.), Takeda Science Foundation (to Y.D.), Naito Foundation (to Y.D.), Sumitomo Foundation (to Y.D.), Novartis Foundation (Japan) for the Promotion of Science (to Y.D.), Foundation for Promotion of Cancer Research in Japan (to Y.D.), and by scholarship support from The Tokushukai Scholarship Foundation (to K.T.).

**Data Availability Statement:** Data presented in this study are available on request from the corresponding author.

**Conflicts of Interest:** The authors declare no conflict of interest.

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
