# Peer review of "Helical Foldamers and Stapled Peptides as New Modalities in Drug Discovery: Modulators of Protein-Protein Interactions"

_processes, doi:10.3390/pr10050924_

Round 1

Reviewer 1 Report

Mizawa, Demizu and coworkers propose a mini-review focusing on peptide foldamers (mainly helix mimetics) as new therapeutics able to inhibit protein-protein interactions (PPIs). Such field is of great importance since resistance mechanism against small molecules and selectivity issues frequently arise. To overcome these drawbacks, many efforts were attempted to target PPIs surfaces. Nevertheless, targeting spatially extended protein interfaces is highly challenging using small molecules, and often imposes a counterproductive tradeoff between specificity and potency. In this context, potent and selective foldamers, resistant to proteolysis, are promising tools in drug discovery.

The authors have made great efforts to provide a comprehensive overview on the topic. It is important but quite an arduous task to clearly delineate the contours of the manuscript and to report well-chosen references providing to the reader a global overview of what is being done in the area. Before acceptance, I suggest modifications (major and minor points, see below) to strengthen the manuscript. Also, in my opinion, some illustrations should be added in the different parts of the review to help the reader especially for non-specialist.

-I agree that helices are by far the most widely found structural motif among foldamers and that helices are preponderant structures at the protein-protein interfaces (Jochim, A. L.; Arora, P. S. ACS Chem. Biol. 2010, 5, 919–923) but more and more examples of β-strand/sheet and ribbon mimetics are now found the literature. These latter are not described in this review, thus I propose the authors to adapt their title, e.g ‘Helical or Helix foldamers […]”. The authors chose to focus on “biotic” foldamers whose building blocks are separated by amide or urea groups, including β-, γ-peptides, oligoureas. peptoids and aromatique/terphenyl foldamers. In the same class, the existence of other families of scaffolds such as  δ- and ε-peptides, aza/aza-b3-peptides, pyrrolinones, α-aminoxy-peptides and sugar-based peptides could have been mentioned in introduction.

-The first paragraph of the introduction must be strengthened. In my opinion, it is important to complete the definition of foldamer by Gellman by Hill’s definition (Please cite: Hill D.J. et al., Chem. Rev. 2001, 101, 3893 – 4011). Hills define a foldamer as “any oligomer that folds into a conformationally ordered state in solution”. Also, the pioneering works of D. Seebach and coworkers are lacking (Seebach D. et al. Chem. Biodiversity 2004, 1, 1111 – 1239, and references therein). Peptides, DNA etc. are not foldamers since they are not artificial molecules.

Then, provide general references for each fields in the following sentences (p1, l28): ‘materials chemistry (add refs), polymer chemistry (add refs)’ etc. Please, also mention the use of foldamers in organocatalysis (two recent reviews: Girvin ZC et al. J Am Chem Soc. 2020;142(41):17211-17223 and Legrand et al. Catalysts 2020, 10(6), 700) and cite recent articles from other groups, on foldamers as CPP: e.g. Vezenkov LL et al. Chembiochem. 2017;18(21):2110-2114, Bornerie et al. Chem Commun (Camb). 2021 Feb 15;57(12):1458-1461, etc. ; or as antimicrobials : e.g.  Bhaumik KN et al. Mol Syst Des Eng. 2021;7(1):21-33, Bonnel C et al. J Med Chem. 2020;63(17):9168-9180.

-The characteristic of the α-helix at the beginning of the first paragraph, α-peptide, should be moved in the introduction.

-Indeed, stapling is one of the most established strategies to date for stabilizing α-helical structures in short peptide sequences and for inhibiting PPIs, but stapled peptides are definitively not foldamers but constrained peptides (i.e. cyclic peptides). There are lots of examples of potent stapled peptides in the literature that deserve a separate and specific review. In my opinion, the paragraph on stapled peptides must be removed from the present manuscript. Nevertheless, stapled peptides and PROTACs can be briefly mentioned in the introduction as alternatives.

-Please adapt l203: […] side-chain on the β2 or/and β3 position from the carboxylic acid. Others recent refs on cyclic and bicyclic β-amino acids: Kiss L and al. Amino Acids. 2017 (9):1441-1455; Milbeo P et al. Acc Chem Res. 2021 Feb 2;54(3):685-696.

- At the beginning of the  g-peptide section, the references are quite old and some recent achievements are lacking. Please, cite this recent review (Legrand B. et al. Chempluschem. 2021;86(4):629-645) which exhaustively recapitulate the different g-building blocks and g-peptide shapes with various applications in drug discovery such as AMPs, vectorization and also an application which fall into the scope of the present manuscript by using a g-peptide 9-helix to target amyloid oligomers aggregation (please, add references: Mathieu L. et al.. Angew Chem Int Ed Engl. 2013;52(23):6006-10, and Kaffy J et al. Chemistry. 2020;26(64):14612-14622).

-In the paragraph on peptoids, publications by the group of Taillefumier are missing. e.g. Szekely T et al. J Med Chem. 2018;61(21):9568-9582, Shyam R et al. ChemMedChem. 2018, 13(15):1513-1516, etc.

Minor points :

P1, abstract : Foldamers […] that can target […].

TOC: remove side chain stappling and other representative building blocks.

Fig1: Please the improve the quality of the figure.

Reviewer 2 Report

The authors introduced research on peptide-based foldamers composed of several building blocks and their applications as proton pump inhibitors. The mini-review is demanding and provides a useful summary of the structure and stability aspects of various types of foldamers. Since their potential drug delivery applications in the focus of the review, their safe human application is a key issue. Therefore, I propose the addition of a subchapter about their safety characterization.

Round 2

Reviewer 1 Report

The authors have strongly improved the first version of their manuscript by clearly indicating that the mini-review focuses on helix structures: foldamers AND stapled peptides. Importantly, even if PPI inhibitors is the main topic, other applications are mentioned such as antimicrobials, vectors and catalysts. Also, the readers will appreciate the numerous and various references allowing them to have an updated overview of the state of art.  

Minor points:

l30: “a conformationally” instead of “aconformationally”

l486: “kondrotas” instead of “kondrptas”
